# Examining the Coordination Between Green Finance and Green Economy Aiming for Sustainable Development: A Case Study of China

**Nana Liu [1], Chuanzhe Liu [1,\*], Yufei Xia [2], Yi Ren [1] and Jinzhi Liang [1]**

[1] School of Economics and Management, China University of Mining and Technology, Xuzhou 221116, China; liunana1004@cumt.edu.cn (N.L.); ts18070051a31@cumt.edu.cn (Y.R.); ts19070045p21@cumt.edu.cn (J.L.)

[2] Business School, Jiangsu Normal University, Xuzhou 221116, China; 6020180093@jsnu.edu.cn

\* Correspondence: rdean@cumt.edu.cn; Tel.: +86-516-83590168

**Abstract:** Green finance (GF) regards social responsibility and environmental protection interests as the core of development and has become a new growth point and a new engine for promoting the development of the green economy (GE). To more accurately grasp the coordination between GF and the GE, the selection of appropriate indicators and feasible methods is worth exploring. Aiming at sustainable development by evaluating the coupling coordination between GF and the GE by means of a comprehensive index system and an integrated approach, this study establishes a coupling coordination degree model based on panel data of 30 Chinese provinces over the period 2007–2016. Furthermore, it evaluates the spatial distribution difference and dynamic evolution trend of the coordination by introducing global/local spatial autocorrelation, a space Markov chain, and a local indicators of spatial association (LISA) Markov chain. According to the research results, the coupling coordination degrees of the provinces exhibit gradual upward trends, and most regions in China are in a barely coordinated state at present. The coordination degree of GF and the GE shows strong spatial dependence overall, and partially presents the characteristics of "high-high (HH)" and "low-low (LL)" clustering patterns. The forecast results show that the future coordination of GF and the GE will remain stable and be affected by the coordinated development of surrounding areas.

**Keywords:** green finance; green economy; coupling coordination; space Markov chain; local indicators of spatial association (LISA) Markov chain

## 1. Introduction

Environmental problems such as ecological imbalance, resource exhaustion, and environmental pollution have become global economic and political problems because they are closely related to social development and human survival [1,2]. The efficient green economy (GE), which boasts low energy consumption, low pollution, and low emissions, has become a necessary choice and direction for China's economic advancement and a channel to help developing countries achieve sustainable development [3]. The development of the GE is inseparable from the support of green finance (GF). On the basis of traditional finance, GF regards social responsibility and environmental protection interests as the core of development, and has become a new growth point and a new engine for promoting the development of the GE [4–6]. In August 2016, seven ministries and commissions of China issued Guidelines for Establishing the Green Financial System which explicitly proposed to promote green economic transformation through constructing a GF system. In practice, all kinds of GF investment are also increasing. According to public data, the balance of the green credit of 21 major domestic banks reached ¥8.22 trillion by the end of June 2017.

GF is an emerging concept. Early research focused on the theoretical analysis of the concept and system structure [7–9]. Soppe (2009) [10] introduced the concept of sustainability into financial practice and financial academic literature. Some researchers have studied profitability in the GF development of financial institutions. Chami et al. (2002) [11] and Scholtens and Dam (2007) [12] revealed that financial institutions who implemented GF and the Equator Principles could gain social recognition and renown, enabling them to successfully carry out financial business and improve financial performance. For GF instruments, Climent and Soriano (2011) [13] investigated and compared the performance between US green mutual funds and other socially responsible investing mutual funds by using a CAPM-based methodology, concluding that green mutual funds had poorer performance than other conventional funds. Based on daily closing prices of the S&P Green Bond index, Pham (2016) [14] identified the fluctuations of the green bond market over the period 2010–2015. Antimiani et al. (2017) [15] developed a computable general equilibrium model of dynamic climate and economy to research how the Green Climate Fund potentially compensated for adaptation and mitigation behaviors under the global climate framework. Cui and Huang (2018) [16] discussed several schemes for raising the public finance of the Green Climate Fund among developed countries, namely, the historical emission responsibility (HR), ability to pay (AP), United Nations (UN) membership dues, Official Development Assistance (ODA), and Global Environment Facility (GEF) approaches. Among these schemes, HR and AP have been widely examined, whereas the remaining three schemes draw lessons from ongoing international financing mechanisms. Environmental pollution liability insurance could be a useful tool to mitigate the problems of environmental risk [17,18].

Recently, scholars have become increasingly interested in the relationship between GF and the GE. Liu et al. (2019) [19] discussed the influence of GF on GE by establishing a super-efficient slack-based model and an index system for 30 Chinese provinces. After discussing the need for greening the financial system and the role of financial governance, Volz (2018) [20] believed that the financial sector would have to play a primary role in this green transformation. However, there has been little discussion about them so far.

Research on the relationship between economic growth and financial development dates back to the eighteenth century, when Smith (1776) [21] studied economic growth and capital accumulation. At the beginning of the twentieth century, Schumpeter (1911) [22] explored the mechanism for banks to promote economic growth. It has been recognized that financial development can facilitate economic growth, and a causality relationship lies between them [23,24]. Other scholars focused on the relationship between financialization and the real economy. After the financial crisis (i.e. 2008–2009), most studies argued that financialization weakened the investment that supported real economic growth [25]. Nevertheless, Shen and Lee (2006) [26] examined the panel data on 46 countries over 1976–2001 and found that financial development and economic growth shared an inverted-U-shaped relationship. Law and Singh (2014) [27] further determined this threshold effect with the aid of a dynamic panel threshold model.

GF evaluation is the foundation and essential scientific basis for the management of GF development. However, due to the lack of clear quantitative standards and statistical data, the research on GF evaluation still faces many difficulties. Scholars mostly carried out case studies on the concept and the economic and environmental benefits of GF, but they have barely quantitatively measured the development level of GF [28]. In addition, the relationship between GF and the GE is rarely discussed. With the aim of more accurately revealing the status and level of GF development in China, this study constructs a quantitative evaluation system from multiple dimensions under the existing conditions and integrated actual development of GF in developing countries. It also discusses the coordination relationship between GF and the GE. This study can provide valuable reference for promoting the development of GF.

While describing and analyzing coordinated development between the past and the present, most existing research has regarded each region as independent, ignoring the fact that regions interacted with each other spatially. Few researchers forecasted the future coordination state, not to mention

investigated the spatial distribution and dynamic evolution of coordination degree considering the spatial spillover effect. It is essential to study the coupling relationship between GF and the GE from a broader viewpoint, instead of a local one. Moreover, it is also necessary to especially study the temporal and spatial variations in the coupling degree and clustering patterns. Aiming to temporally and spatially reveal the coupling coordination between GF and the GE in China, this study evaluated their relationship over the period 2007–2016 by putting forward a coupling coordination degree model on the basis of a physics coupling model. In order to explore the clusters, spatial association, and spatial dynamics for the coupling coordination, the spatial distribution was described and visualized by means of an exploratory spatial data analysis (ESDA) proposed by the authors of [29]. Moreover, the trend of coordination value was forecasted by using a space Markov chain and a local indicators of spatial association (LISA) Markov chain. The study provides a scientific basis for the practice of GF and sustainable economic development in developing countries.

This study includes five sections. In Section 2, index systems for green finance and the green economy are established. This section gives theoretical foundation regarding the mechanism of coordination between GF and the GE. In Section 3, the materials and methods are introduced, specifically, the entropy method, the space Markov chain, the LISA Markov chain and the ESDA are described in detail. In Section 4, the results of coupling coordination, spatial analysis, and spatial dynamics are presented on the basis of the model put forward in Section 3. Finally, conclusions are given in Section 5.

## 2. Theoretical Foundation

### 2.1. Interactive Relationship Between GF and GE

Financial development is closely related to economic growth. GF and the GE are bound to be closely related because they were both proposed and developed to regulate the contradiction between economic growth and environmental protection (Figure 1). Generally, the GE is the foundation of GF, while GF, which provides important support for the GE, is the driving force of GE development. Finance ignores ecosystems, resulting in increasing environmental and social problems. Financial institutions and markets, which influence ecosystems considerably, should care about ecology because ecology assists them to efficiently and effectively play their social and economic roles [12]. GF is an essential part of low carbon green growth for two reasons. First, it connects financial development, economic growth, and environmental improvement [30]. Second, as a new financial pattern combining environmental protection with economic profits, it attaches importance to the controversial issues "green" and "finance" [31].

GF provides support for the GE, which is primarily embodied in four respects. (1) GF lowers the cost of raising funds during the development of green industries mainly by collecting and guiding funds to these industries. In this way, it provides favorable conditions for the development of green industries. (2) GF indirectly raises the cost of high-pollution projects by reducing the costs of green investment and financing, energy conservation, and emission reduction and making environmental risk explicit. In this way, it restrains polluting investments [32]. (3) Investors consider the potential environmental effects of investment and financing decisions; that is, they consider the environmental risks and costs of investment decisions and lay stress on the protection of the ecological environment and the control of environmental pollution. (4) GF can integrate environmental risks with financial risks and effectively work out the problem of market failure by taking advantage of financial risk management techniques and various forces such as market mechanism, government regulation, and social supervision. Besides, it focuses on prevention before a risk event and supervision during the event instead of punishment after the event [33].

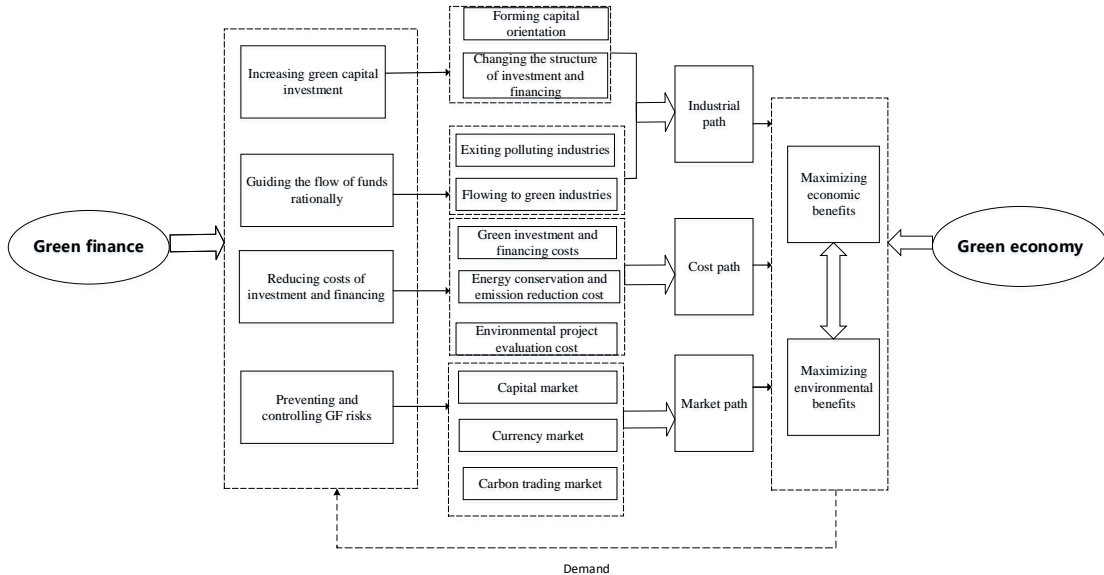

**Figure 1.** The coupling coordination relationship between green finance and green economy.

The GE is the foundation of GF. It is known that traditional finance is a part of traditional economic development, and economic development is the basis of financial development. Similarly, GE development is also the basis of GF development, because without the development of the GE, there will be no demand for funds, and it is impossible to guide the attention to ecological environment through funds. To develop the GE, it is necessary to constrain the development behavior of various economic organizations so that they attach importance to environmental protection in the development process. In contrast, GF intervenes through economic means at the level of financial support and urges economic development institutions to take environmental factors into account in their development behaviors. In this way, the contradiction between environmental protection and economic development can be coordinated. In short, the GE promotes the emergence of GF and serves as the basis for GF development.

*2.2. Index System*

To study the coupling coordination between GF and the GE in China, a comprehensive evaluation system was established in accordance with the existing research [29]. Indicators of GF and the GE were initially selected based on four principles [34,35]: (a) to select the most cited indicators; (b) to cover the contents of GF and the GE comprehensively; (c) to choose the most representative indicators for facilitating data collection, understanding, and multi-collinearity [36]; (d) to follow the policies, opinions, and guidelines promulgated by the governments of different regions, for example, Guiding Opinions on Building a Green Financial System jointly issued by the seven ministries of China on 31st August 2016.

According to different types of financial services, the GF system can be divided into five dimensions: green credit, green securities, green insurance, green investment, and carbon finance. Specifically, regarding the index of green credit, due to the limited disclosure of provincial green credit information by financial institutions, the proportion of interest expenses of high-energy-consumption industries (X1) was introduced as a reverse indicator with reference to [19]. Currently, the industry loan interest rate gap in China is relatively small. In this case, the change in interest expense are mainly related to the scale of loans, indirectly reflecting the key areas of changes in the proportion of loans. Therefore, X1 could reflect the strength of commercial banks to curb the deterioration of resource environment and were negatively correlated with the development of green credit.

The index of green securities, consisting of market value ratios of environmental protection enterprises (X2) and high-energy-consumption industries (X3), mainly reflects the financing level of China's environmental protection industries and high-energy-consumption industries through the

issuance of stocks in the capital market. The six high-energy-consumption industries refer to the chemical raw materials and chemical manufacturing industry, the non-metallic mineral products manufacturing industry, the ferrous metal smelting and rolling processing industry, the non-ferrous metal smelting and rolling processing industry, the petroleum processing coking and nuclear fuel processing industry, and the power and heat production and supply industry. In this study, the environmental protection industries were selected according to the main business of companies, including listed companies, in 30 provinces (municipalities and autonomous regions) except Tibet, Hong Kong, Macao and Taiwan. They include the Beautiful China concept, wind power generation, green energy-saving lighting, sewage treatment, and tail gas treatment. The index of green insurance contains two third-class indicators: the ratio of agricultural insurance (X4) and the loss ratio of agricultural insurance (X5). China did not enforce enterprise environmental liability insurance until the end of 2013, so there is a lack of authoritative statistical data. Since agriculture is an industry that is greatly affected by the natural environment, the scale and compensation rate of agricultural insurance can approximately reflect the development of green insurance. The index of green investment includes two third-class indicators: the ratio of public expenditure on energy conservation and environmental protection (X6) and the ratio of environmental pollution control investment (X7). Considering the small amount of data on China's participation in clean development mechanism (CDM) project transaction volume, the paper does not regard carbon finance as an indicator.

In this study, an effective GE evaluation index system was proposed with reference to the driving forces-pressure-state-impact-response (DPSIR) framework with the consideration of the actual situation of the economy [37] (Figure 2). The DPSIR framework was proposed in the late 1990s as a tool for reporting and analyzing environmental problems [38]. This framework, which can evaluate green development more effectively than the original PSR framework, has been extensively adopted in environment and economy research like Pissourios (2013) [36] and applied by international organizations.

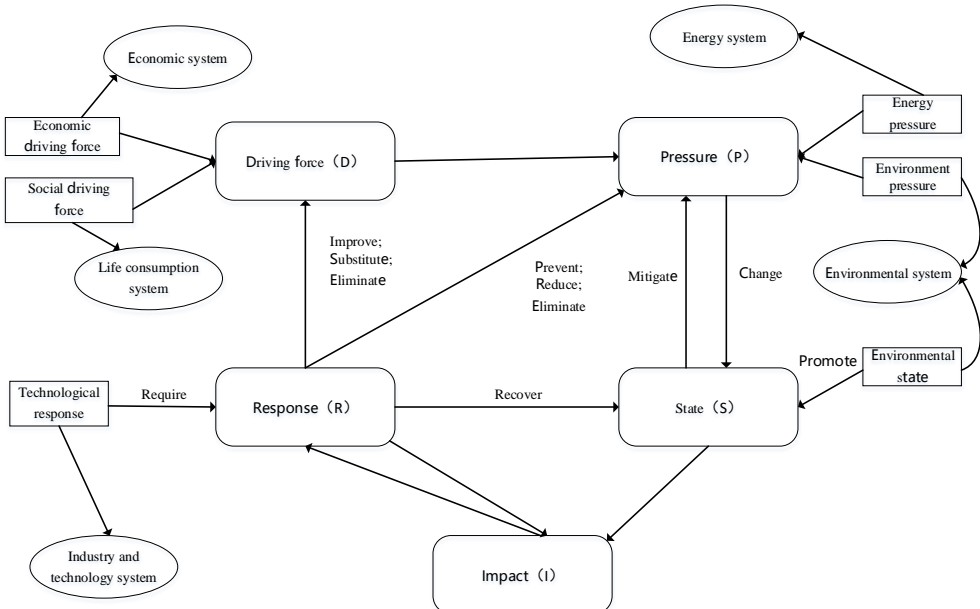

**Figure 2.** A driving force-pressure-state-response (DPSR) framework for the analysis of green economy indicators.

Since the "Impact (I)" factor in the DPSIR model is not only uncertain but also difficult to measure in GE development evaluation, this study modified the model to a driving force-pressure-state-response (DPSR) model with reference to [39]. According to the green development indexes released by the National Development and Reform Commission of China in 2010, the GE evaluation system is divided

into five parts. Among them, driving force (D) is composed of economic development driving forces (D1) and social development driving forces (D2); pressure (P) refers to environmental pressure (P1); state (S) refers to environmental state (S1); and response (R) can be expressed as technological response (R1) and environmental response (R2). The DPSR model for constructing the system of GE development evaluation is given in Table 1.

**Table 1.** Index system for green finance.

| First-Class Indicators | Weights | Second-Class Indicators | Description | Weights |
|---|---|---|---|---|
| Green credit | 0.5 | Proportion of interest expenses in high-energy-consumption industries (X1) | Interest expenses of six high-energy-consumption industries/total interest expenditure of industries | 1 |
| Green securities | 0.25 | Market value ratios of environmental protection enterprises (X2) | Market value of environmental protection enterprises/total market value of A shares | 0.5 |
| | | Market value ratios of high-energy- consumption industries (X3) | Market value of six high-energy-consumption industries/total market value of A shares | 0.5 |
| Green insurance | 0.15 | Ratio of agricultural insurance (X4) | Agricultural insurance expenditure/total insurance expenditure | 0.5 |
| | | Compensation ratio of agricultural insurance (X5) | Agricultural insurance expenditure/agricultural insurance income | 0.5 |
| Green investment | 0.1 | Ratio of public expenditure on energy conservation and environmental protection (X6) | Energy conservation and environmental protection fiscal expenditure/total fiscal expenditure | 0.5 |
| | | Ratio of environmental pollution control investment (X7) | Environmental pollution control investment accounts/GDP | 0.5 |

Data sources: Wind database.

GE can generally be regarded as a method to lower pressure on resources and emissions and meanwhile ensure economic growth and social welfare [40]. In this study, the third-class indicators of GE are chosen according to the growth quality index of the Green Development Indicator System published by the National Development and Reform Commission in 2016. Given data accessibility and multi-collinearity, GDP per capita (D11), consumption level per capita (D12), and GDP growth rate (D13) are chosen among many potential options to quantify economic development driving forces (D1).

The GE has been defined as being low-carbon, resource efficient, and socially inclusive [41]. Social development driving forces (D2) are also important parts of the GE. In this study, living water consumption per capita (D21), the number of buses per 10,000 people (D22), the number of beds in health institutions per 10,000 people (D23), and the urban road area per capita (D24) were chosen as third-class indicators of D2. Environmental pressure (P1) is affected by many factors such as serious waste of water resources, environmental pollutants, and the increasing exploitation of mineral resources. Energy consumption per capita (P11), industrial wastewater discharge (P12), solid waste discharge (P13), and SO2 emissions (P14) were selected as the representative indicators of P1. However, this study differs slightly with that of [19] for the addition of energy consumption per capita (P11) because it contributes to carbon emissions. Environmental state (S) is the most direct manifestation of the level of green development, and the better the environmental state the higher the level of green development. The park green area per capita (S11), the garden green area per 10,000 people (S12),

the built-up area green coverage (S13), and the cover rate of forest (S14) are often selected as the representative indicators of S1.

Resource efficiency mainly aims to promote the utilization of natural resources in the production value chain and decrease emissions and waste in the environment through technological innovations [42]. Third-class indicators of technological response (R1), the development of tertiary industry, and the investment on research play a crucial role in reducing the consumption of non-renewable resources and the emissions of pollutants. The proportion of science and technology expenditure is accounted for in fiscal expenditure (R11), the proportion of tertiary industry is accounted for in GDP (R12), and the number of patents (R13) were chosen to measure the level of technological development. The third-class indicators of environmental response, urban sewage daily treatment capacity (R21), the comprehensive utilization rate of industrial solid waste (R22), and the harmless treatment rate of domestic garbage (R23) were selected to measure the human effort to reduce pollution with reference to the environmental development indicators given by Green Development Indicator System and the research results of [43].

Hence, the final comprehensive GF index system is composed of four first-class indicators and seven second-class indicators (Table 1), while the final comprehensive GE index system is comprised of five first-class indicators, six second-class indicators and twenty-one third-class indicators (Table 2).

**Table 2.** Index system for green economy.

| System | First-Class Indicator | Second-Class Indicator | Third-Class Indicator | Weight |
|---|---|---|---|---|
| GE | Driving forces (D) | Economic development driving forces (D1) | GDP per capita (yuan) (D11) | 0.0582 |
| | | | Consumption level per capita (yuan) (D12) | 0.0791 |
| | | | GDP growth rate (%) (D13) | 0.0057 |
| | | Social development driving forces (D2) | Living water consumption per capita (liter) (D21) | 0.0385 |
| | | | Number of buses per 10,000 people (D22) | 0.0523 |
| | | | Number of beds in health institutions per 10,000 people (D23) | 0.0345 |
| | | | Urban road area per capita (m$^2$) (D24) | 0.0301 |
| | Pressure (P) | Environmental pressure (P1) | Energy consumption per capita (tons) (P11) | 0.0068 |
| | | | Industrial wastewater discharge (10,000 tons) (P12) | 0.0168 |
| | | | Solid waste discharge (10,000 tons) (P13) | 0.0087 |
| | | | SO$_2$ emissions (tons) (P14) | 0.0194 |
| | State (S) | Environmental state (S1) | Park green area per capita (m$^2$) (S11) | 0.0342 |
| | | | Garden green area per 10,000 people (10000 m$^2$) (S12) | 0.0845 |
| | | | Built-up area green coverage (%) (S13) | 0.0145 |
| | | | Cover rate of forest (%) (S14) | 0.0585 |
| | Response (R) | Technological response (R1) | Proportion science and technology expenditure accounts for in fiscal expenditure (%) (R11) | 0.0535 |
| | | | Proportion tertiary industry accounts for in GDP (%) (R12) | 0.0482 |
| | | | Number of patents (R13) | 0.2240 |
| | | Environmental response (R2) | Urban sewage daily treatment capacity (10,000 m$^3$) (R21) | 0.0886 |
| | | | Comprehensive utilization rate of industrial solid waste (%) (R22) | 0.0272 |
| | | | Harmless treatment rate of domestic garbage (%) (R23) | 0.0167 |

Data sources: Wind database.

## 3. Materials and Methods

### 3.1. Data Source and Pre-Processing

The annual data of the GF and GE system in 30 Chinese provinces over the period 2007–2016 were taken online from the *China Statistical Yearbook* (2008–2017), *China City Statistical Yearbook* (2008–2017), *China Energy Statistical Yearbook* (2008–2017), *China Environmental Statistics Yearbook* (2008–2017), and *China Environmental Quality Report* (2008–2017). For the missing data, the post-evaluation was calculated based on the average annual growth rate. Although Industrial Bank Co., Ltd. (Fujian, China) had begun its GF practice in 2005, *Opinions on Implementing Environmental Policy and Regulations to Prevent Credit Risk*, jointly issued by the China Banking Regulatory Commission, the People's Bank of China, and the State Environmental Protection Administration in 2007, truly heralded the start of China's GF practice. Hence, the data obtained in this study started from 2007. The data on green finance and the green economy for 2016 are presented in Appendices A and B, respectively.

For the consistency and comparability of empirical results, the dimensionless values of basic data were obtained in light of the max–min treatment method in [28], as shown in Equation (1):

$$x'_{ijt} = \begin{cases} \frac{x_{ijt}-min(x_j)}{max(x_j)-min(x_j)}, \text{Positive indicator} \\ \frac{max(x_j)-x_{ijt}}{max(x_j)-min(x_j)}, \text{Negative indicator} \\ \frac{1}{1+|x_{ijt}-M|}, \text{Moderate indicator} \end{cases} \tag{1}$$

where $i$ is the province; $j$ represents the indicator; $t$ is the year; $x_{ijt}$ is the value of original data; $max(x_j)$ and $min(x_j)$ are the maximum and minimum values of indicator $j$ in all the years studied, respectively; and $M$ is the mean value of $x_{ijt}$.

### 3.2. Methods

#### 3.2.1. Entropy Method

The weighting of each indicator in the indicator system is a necessary part of coupling coordination between GF and the GE. The entropy method has been extensively adopted to determine the weights of environmental and economic indicators [44,45]. Supposing E stand for the indexes of the GE subsystem, and F denote the indexes of GF subsystem, then Equations (2) and (3) exist:

$$f(E) = \sum_{e=1}^{p} w_e E'_e' \tag{2}$$

$$f(F) = \sum_{f=1}^{q} w_f F'_f' \tag{3}$$

where $f(E)$ and $f(F)$ are the integration values of GE and GF, respectively; $E'_e$ *and* $F'_f$, which can be calculated by Equation (1), are the standardized values of $f(E)$ and $f(F)$, respectively; and $w_e$ and $w_f$ are the weights.

The entropy weight of each index was calculated based on its degree of variation [46]. The formula used was as follows:

Calculating the proportion of indicator $j$ in the year $i$:

$$p_{ij} = x_{ij} / \sum_{i=1}^{m} x_{ij}. \tag{4}$$

Calculating the information entropy $e_j$ of indicator:

$$e_j = -1/ln_m \sum_{i=1}^{m} p_{ij} \times lnp_{ij}, \tag{5}$$

where $k = 1/ln_m$.

Calculating the entropy redundancy $d_j$ of indicator:

$$d_j = 1 - e_j. \tag{6}$$

The entropy redundancy $d_j$ is negatively correlated with the information entropy $e_j$, that is, $d_j$ is greater when the value of $e_j$ is smaller.

Calculating the weight $W_j$ of indicator:

$$W_j = f_j / \sum_{j=1}^{n} f_j \left(0 \leq W_j \leq 1\right). \tag{7}$$

### 3.2.2. Coupling Coordination Degree Model

Based on research ideas and model construction methods in physics, the dynamic changes and evolution process of coupling coordination development of GF and GE were analyzed in light of [47]. This paper introduces a coupling coordination degree model which can be expressed by the following formulas:

$$C = \sqrt{\frac{f(E) \times f(F)}{(f(E) + f(F))/2}}, \tag{8}$$

$$T = af(E) + bf(F), \tag{9}$$

$$D = (C \times T)^{\frac{1}{2}}, \tag{10}$$

where $C$ is the degree of coupling; $D$ is the coupling coordination degree between GE and GF, and $D \in [0,1]$; T is the index for comprehensively evaluating the coordinated development; and a and b are the weights of GE and GF for sustainable economic development, respectively. According to the research of REN et al. (2011) [48] and the actual development of China, the GE and GF systems make equally important contributions to the coupling coordination degree. Hence, the values of *a* and *b* are determined to be the same, i.e., $a = b = 0.5$.

Drawing on the existing research results and studies on the coupling coordination degree [47,49], the coordination values were divided into three major classes and six subclasses, as presented in Table 3.

**Table 3.** The classification of coordination degree between green finance and green economy.

| Classes | | Subclasses | | Types |
|---|---|---|---|---|
| Balanced development (Acceptable interval) | 0.8–1 | Superior balanced development | $0 \leq |f(x) - e(y)| \leq 0.1$ | Superior balanced development with green finance and green economy |
| | | | $f(x) - e(y) > 0.1$ | Superior balanced development with green economy lagged |
| | | | $e(y) - f(x) > 0.1$ | Superior balanced development with green finance lagged |
| | 0.6–0.8 | Favorably balanced development | $0 \leq |f(x) - e(y)| \leq 0.1$ | Favorably balanced development with green finance and green economy |
| | | | $f(x) - e(y) > 0.1$ | Favorably balanced development with green economy lagged |
| | | | $e(y) - f(x) > 0.1$ | Favorably balanced development with green finance lagged |

**Table 3.** *Cont.*

| Classes | Subclasses | | Types |
|---|---|---|---|
| Transitional development (Transitional interval) | 0.5–06 | Barely balanced development | |
| | | $0 \leq |f(x) - e(y)| \leq 0.1$ | Barely balanced development with green finance and green economy |
| | | $f(x) - e(y) > 0.1$ | Barely balanced development with green economy lagged |
| | | $e(y) - f(x) > 0.1$ | Barely balanced development with green finance lagged |
| | 0.4–0.5 | Slightly unbalanced development | |
| | | $0 \leq |f(x) - e(y)| \leq 0.1$ | Slightly unbalanced development with green finance and green economy |
| | | $f(x) - e(y) > 0.1$ | Slightly unbalanced development with green economy hindered |
| | | $e(y) - f(x) > 0.1$ | Slightly unbalanced development with green finance hindered |
| Unbalanced development (Unacceptable interval) | 0.2–0.4 | Moderately unbalanced development | |
| | | $0 \leq |f(x) - e(y)| \leq 0.1$ | Moderately unbalanced development with green finance and green economy |
| | | $f(x) - e(y) > 0.1$ | Moderately unbalanced development with green economy hindered |
| | | $e(y) - f(x) > 0.1$ | Moderately unbalanced development with green finance hindered |
| | 0–0.2 | Seriously unbalanced development | |
| | | $0 \leq |f(x) - e(y)| \leq 0.1$ | Seriously unbalanced development with green finance and green economy |
| | | $f(x) - e(y) > 0.1$ | Seriously unbalanced development with green economy hindered |
| | | $e(y) - f(x) > 0.1$ | Seriously unbalanced development with green finance hindered |

### 3.2.3. Global and Local Spatial Autocorrelation

As the earliest method for global clustering tests, Moran's I was used to test whether adjacent regions within the study area were related or independent. The definition is expressed in vector form:

$$I = \frac{\sum_{i=1}^{m} \sum_{i'=1}^{m} \varnothing_{ii'} D_i - \overline{D} D_{i'} - \overline{D}}{S^2 \sum_{i=1}^{m} \sum_{i'=1}^{m} \varnothing_{ii'}} ,$$

(11)

where $i$ and $i'$ are provinces; $m$ is the total number of regions; and $\varnothing_{ii'}$ is the spatial weight (e.g., for provinces i and $i'$, if i and $i'$ are adjacent, then $\varnothing_{ii'} = 1$; if they are not adjacent, then $\varnothing_{ii'} = 0$.); $D_i$ and $D_{i'}$ are the coordination values of $i$ and $i'$, $\overline{D} = \frac{1}{m} \sum_{i=1}^{m} D_{i'}$, $S^2 = \frac{1}{m} \sum_{i=1}^{m} D_i - \overline{D}^2$.

Moran's I can also be taken as a correlation coefficient between the observed value and its spatial lag. The formula can be expressed as:

$$D_{i,-1} = \sum_{i'} \varnothing_{ii'} D_{ii'} \Big/ \sum_{i'} \varnothing_{ii'}$$

(12)

Moran's I $\in [0, 1]$ means that the attribute values of different regions are spatially dependent, which is a positive correlation. If the value equals 1, it suggests a completely positive correlation. If the value approximates 0, it means that the attribute values of different regions are randomly distributed in space. A value approximating −1 suggests that the attribute values are negatively correlated.

The local Moran's I is a measure of the degree of association between $i$ and an adjacent region $i'$, and its formula is:

$$I_i = \frac{D_i - \overline{D}}{S^2} \sum_{i' \neq i} \varnothing_{ii'} D_{i'} - \overline{D}.$$

(13)

In this study, the local spatial correlation was described by a Moran scatter plot, and the four quadrants were classified into high-high (HH), low-high (LH), low-low (LL), and high-low (HL) in accordance with the variation degree between the provinces and their adjacent regions.

### 3.2.4. Space Markov Chain and LISA Markov Chain

As a combination of a weighted Markov chain and space lag, the space Markov chain is employed to discuss the dynamic spatial and temporal evolution trends of an index. The traditional $k \times k$ Markov matrix is decomposed into $k$ $k \times k$ conditional transfer probability matrices $p_k$ under the condition of space lag type of region $i$ in the initial year $t_0$ (let the set of spatial lag types be K; there are $k$ types). For the $k$th conditional matrix, under the condition of spatial lag type $k$ of the region in the year $t$, $P_{ab|k}(t, t+1)$ denotes the conditional transfer probability that the region belongs to type a in the year $t$ and becomes type b in the year $t + 1$, as shown in Equations (14) and (15).

$$P_k = \left\{ P_{ab|k}, a, b, k \epsilon \mathrm{K} \right\} \tag{14}$$

$$P_{ab|k}(t, t+1) = P\{X(t+1) = b \big| Xt = a, \varnothing X(t) = k\}, a, b, k \epsilon \mathrm{K} \tag{15}$$

With the aid of the PySAL package in Python software, the specific procedure of space Markov chain prediction method is as follows:

Step 1: A Rook space weight matrix based on first-order adjacency relation was generated by GeoDa software and denoted by $\varnothing$. The coordination value D and $\varnothing$ were imported into Python to standardize $\varnothing$.

Step 2: The mean values and mean square errors of samples, based on which the clustering pattern state values $a$ were classified into 5 levels, namely, S1, S2, S3, S4 and S5. The formulas were:

$$\overline{D} = \frac{1}{m * T} \sum_{t=1}^{T} \sum_{i=1}^{m} D_{it} \quad i = 1, 2 \ldots, m; t = 1, 2, \ldots, T \tag{16}$$

$$S = \sqrt{\frac{1}{m * T} \sum_{t=1}^{T} \sum_{i=1}^{m} D_{it} - \overline{D}^2} \, i = 1, 2 \ldots, m; t = 1, 2, \ldots, T. \tag{17}$$

where $\overline{D}$ is the mean value; S is the mean square error; $D_{it}$ is the coordination degree of the ith province in the year $t$; m is the number of provinces; and T is the total time.

Step 3: The $l$-order correlation coefficient was calculated.

$$r_l = \frac{\sum_{t=1}^{T-l} D_{it} - \overline{D}_i D_{t+l} - \overline{D}_i}{\sqrt{\sum_{t=1}^{T-l} D_{it} - \overline{D}_i^2 \sum_{t=1}^{T-l} D_{it+l} - \overline{D}_i^2}}, \quad i = 1, 2, \ldots, m; t = 1, 2, \ldots, T. \tag{18}$$

where $D_{it}$ is the coordination degree of the $i$th province in the year $t$; $\overline{D}_i$ is the mean coordination degree of the $i$th province; $t$ is the year; and $l$ is the order.

After autocorrelation coefficients of each order were calculated, standardization was performed, and the standardized result $\Phi_l$ was taken as the weight of Markov chains with various step sizes. The standardized processing was:

$$\Phi_l = |r_l| / \sum_{l=1}^{L} |r_l|. \tag{19}$$

Step 4: With the coordination degree of previous moments taken as the initial state, the transfer probability of the state corresponding to the coordination degree $p_a{}^k$ in the year $t$ was calculated. In $p_a{}^k$, $a$ is the state, $a \in \mathrm{I}$; k is the step size, k $= 1, 2 \cdots, k_0$.

Step 5: Prediction probabilities in the same state were weighted, and the results were taken as the prediction probability in that state.

$$P_a = \sum \Phi_l p_a{}^k \ a = 1, 2, \ldots, A. \tag{20}$$

The LISA Markov chain provides the temporal evolution of these spatial clustering patterns. In the four quadrants of the Moran scatter plot, a total of 16 transfer paths can be generated. They are defined as follows:

Condition 1: The total number of areas in clustering pattern c in the year $t$ is $S_c$;

Condition 2: The total number of areas that satisfy condition 1 and develop into clustering pattern d in the year $t+1$ is $G_d$.

Then:

$$P_{cd}(t, t+1) = P\{U(t+1) = d | U(t) = c\} = \frac{G_d}{G_c}. \tag{21}$$

The rest was deduced in the same manner, and the LISA Markov one-step state transfer probability matrix was obtained, as displayed in Table 4. The main diagonal is the probability of maintaining the original clustering pattern, while the other 12 transfer paths are divided into inward diffusion or outward diffusion. Of the two, inward diffusion, including the transfer path from "LH clustering" to "HH clustering" or "HL clustering", refers to the case where the observed value was lower than the average in the current year and became higher than the average in the next year. Outward diffusion, including the transfer path from "HL clustering" to "HH clustering" or "LH clustering", refers to the case where the observed value was lower than the spatial lag value in the current year and became higher than the spatial lag value in the next year. Besides, the other 12 transfer paths were divided into the saturation type and the substitution type. Of the two, the saturation type, including the transfer path from "LL clustering" to "HH clustering", refers to the case where the observed value was greater than both the mean and spatial lag values in the next year. The substitution type refers to the situation where the observed value was inconsistent with the mean and spatial lag values in the next year.

**Table 4.** Sixteen transfer paths of LISA Markov chain.

| Mode | High-High (HH) | Low-High (LH) | Low-Low (LL) | High-Low (HL) |
|------|----------------|---------------|--------------|---------------|
| HH | $P_{11}$ | $P_{12}$ | $P_{13}$ | $P_{14}$ |
| LH | $P_{21}$ | $P_{22}$ | $P_{23}$ | $P_{24}$ |
| LL | $P_{31}$ | $P_{32}$ | $P_{33}$ | $P_{34}$ |
| HL | $P_{41}$ | $P_{42}$ | $P_{43}$ | $P_{44}$ |

## 4. Results and Discussion

### 4.1. Results of Overall Level of Subsystems

Scientific and reasonable weights of indicators directly determine the objectivity and fairness of evaluation results. Since the GE system contains many indicators, including 21 basic indicators, it is difficult to assign accurate weights to the indicators by using subjective methods. In this study, different weights were assigned to GE indicators by means of the entropy weight method. The results are given in Table 2. Among the second-class indicators, technological response (R1) accounted for the greatest weight (32.57%), followed by environmental state (19.17%), social development driving forces (15.53%), and economic development driving forces (14.31%). Among the 21 basic indicators, the number of patents (22.4%), urban sewage daily treatment capacity (8.86%), the garden green area per 10,000 people (8.45%), and the consumption level per capita (7.91%) accounted for 47.62% of the total weight, so they were the key factors influencing development of GE. From these results, it can be concluded that environmental state and response play a more significant role in determining GE development and thus deserve stronger protection in future attempts to realize a sustainable economy.

The weight results of GE index system are exhibited in Table 1. In terms of the weights of second-class indicators, the differences of various GF dimensions were objectively measured according to the proportion of assets in various financial fields. Referring to China Green Finance Report written by Li et al., 50% of weight was assigned to green credit, as it is the most influential component of GF, while 25%, 15%, and 10% were assigned to securities, insurance, and investment, respectively, as they exert a relatively small influence. Average weighting was mainly applied to the third-class indicators because third-class indicators under the same second-class indicator system are relatively independent.

### 4.2. Dynamic Analysis of Coupling Coordination between GF and GE

The coupling coordination values between green finance and green economy for 30 provinces are shown in Appendix C. The 30 Chinese provinces are classified into four regions: East, Central, West, and Northeast, in accordance with the location division standard of China Statistical Yearbook. From Figure 3, it can be observed that the coordination states of the four regions were continuously optimized and the coordination values increased gradually. Specifically, the Eastern region boasted the highest level of coordinated development of GF and the GE, reaching 0.60 in 2014 and realizing the transformation from barely balanced development to favorably balanced development. At present, the coordination values of GF and the GE in the central and northeastern regions lie between 0.5 and 0.6, and they are still in the phase of barely balanced development. The coordinated development of GF and GE in the western region is relatively backward and remains in the phase of slightly unbalanced development.

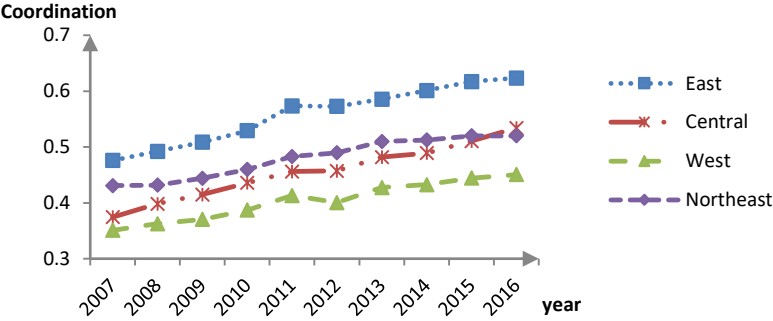

**Figure 3.** Coordination between GF and the GE in the four major regions of China during the period 2007–2016.

The spatial distribution of coordinated development of GF and the GE was obtained based on GIS, so as to more intuitively reflect its spatial pattern evolution. The results are illustrated in Figures 4–6. (1) In 2007, no province reached the phase of balanced development (Figure 4). Only five provinces (Beijing, Guangdong, Jiangsu, Shanghai and Zhejiang) reached the phase of barely balanced development. Nine provinces (Tianjin, Fujian, Hainan, Heilongjiang, Hunan, Jilin, Shaanxi, Shandong, and Xinjiang) were in the phase of slightly unbalanced development, and the rest sixteen provinces were in the phase of moderately unbalanced development. (2) After five years of development, we can see from Figure 5 by 2011, the coordinated development values of four provinces (Guangdong, Jiangsu, Shanghai, and Zhejiang) exceeded 0.6, reaching the phase of favorably balanced development. Eight provinces (Beijing, Tianjin, Chongqing, Fujian, Hainan, Heilongjiang and Shandong) were in the phase of barely balanced development. The values of four provinces (Gansu, Guizhou, Qinghai, and Yunnan) ranged from 0.2 to 0.4 and remained in the phase of moderately unbalanced development. The values of most provinces were still in the phase of slightly unbalanced development.

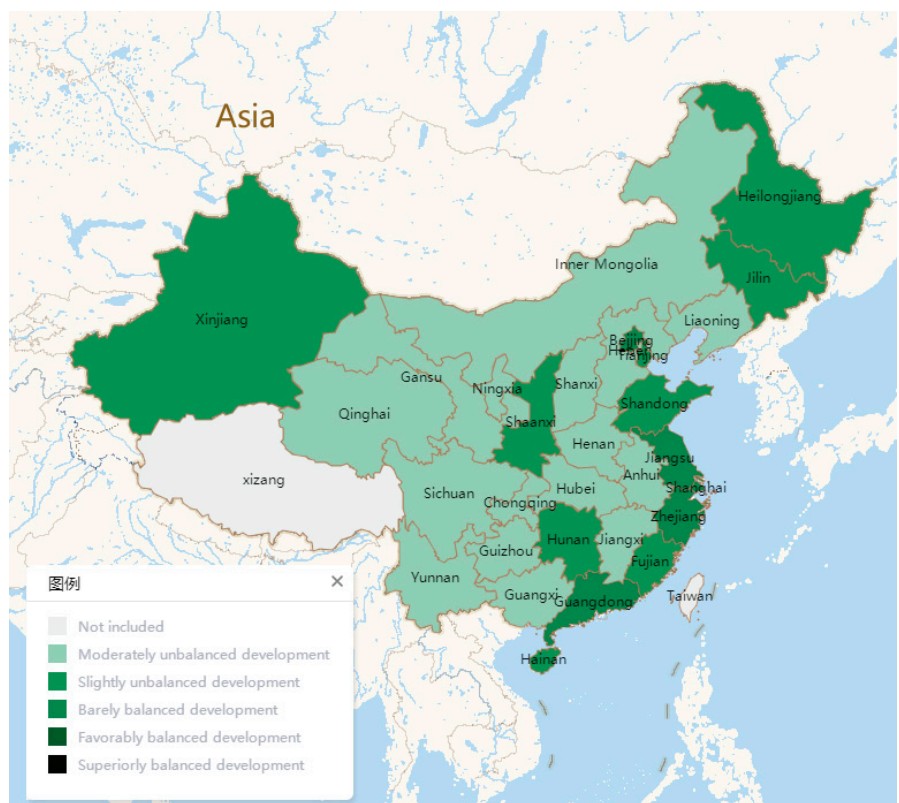

**Figure 4.** Spatial distribution of coupling coordination values in China in 2007.

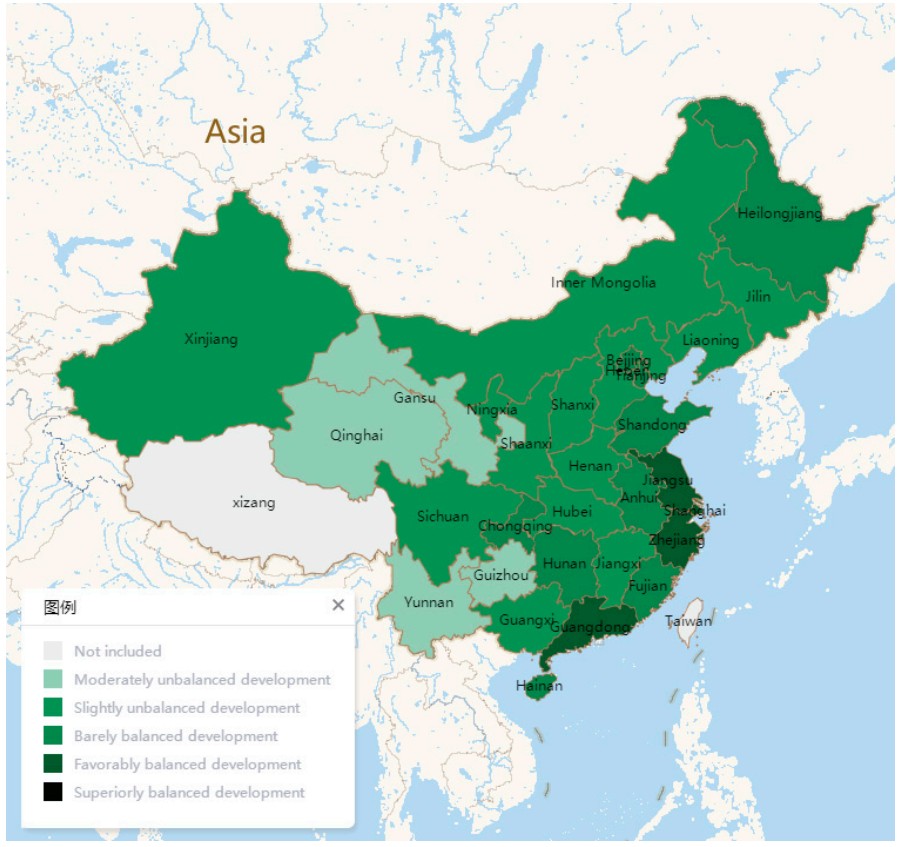

**Figure 5.** Spatial distribution of coupling coordination values in China in 2011.

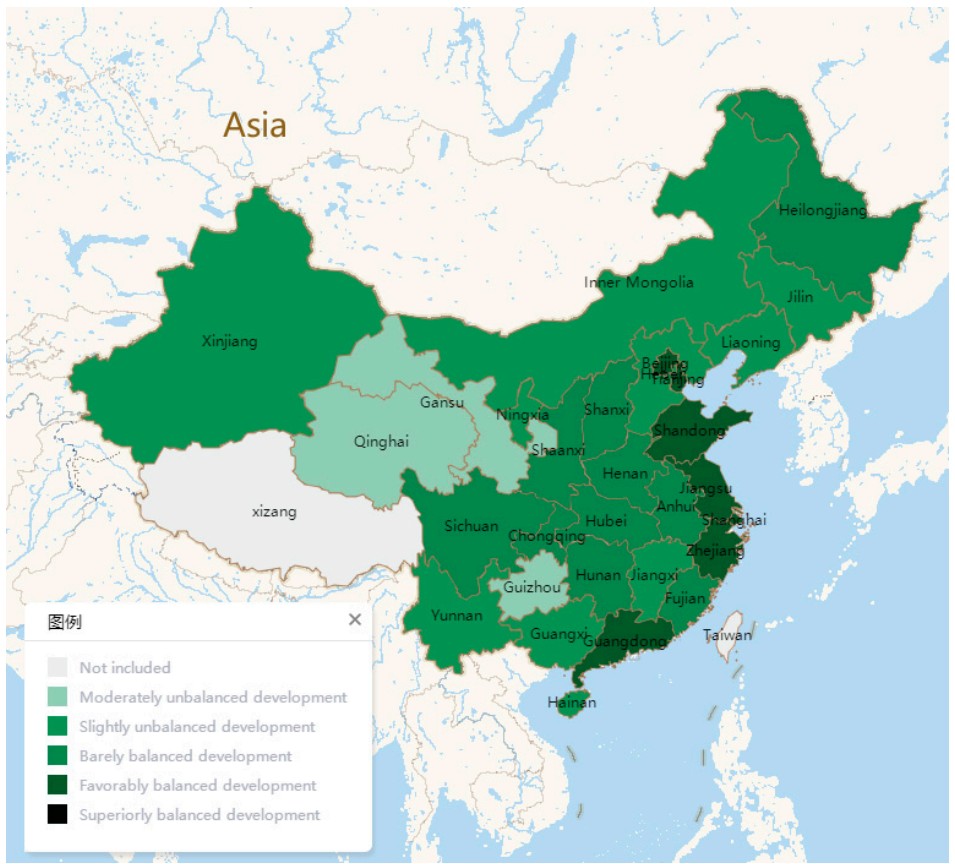

**Figure 6.** Spatial distribution of coupling coordination values in China in 2016.

(3) Figure 6 show that by 2016, seven provinces (Beijing, Tianjin, Guangdong, Jiangsu, Shandong, Shanghai, and Zhejiang) reached the phase of favorably balanced development. Among them, many provinces, such as Beijing, Guangdong, Jiangsu, and Zhejiang, boasted coordinated development values of over 0.7, probably because, after years of development, the GF product system in the Bank of Beijing has operated on a large scale. By the end of 2016, the bank's green loan balance exceeded ¥50 billion, and the development of GF and the GE has been excellently combined. Shanghai is vigorously developing its green energy conservation and environmental protection industry, and it recognized 29 projects of environmental protection, power saving, and water saving and 224 green energy enterprises, and gradually increased investment in green industry. Tianjin issued the Implementation Plan of Controlling Greenhouse Gas Emission in Tianjin during the 13th Five-Year Plan, and it was also actively developing its green energy industry and reducing total coal consumption. The coupling development degree of provinces such as Zhejiang and Jiangsu exceeded 0.6, indicating the obvious synergistic promotion effect between the development of GF and the GE in these provinces.

*4.3. Spatial Dependency Characteristics of Coordination Between GF and the GE*

With the aid of GeoDa software, a Queen spatial weight matrix based on first-order adjacency relation was generated, and the Moran's I value of the coordination degrees between GF and the GE systems of 30 provinces from 2007 to 2016 were calculated (Table 5). At the significance level of 0.05, all the global Moran's I on the basis of the first-order Queen adjacency matrix passed the significance test over the period 2007–2016, with all the values greater than 0. This shows that the difference in coordination values of GF and the GE between Chinese provinces was not randomly distributed in space, but positively correlated. The coordination degree showed strong spatial dependence on the whole. The spatial external correlation and spillover effect can reasonably account for the clustering phenomenon because the effect directly leads to spatial dependence.

**Table 5.** The Moran's I value of coordination over the period 2007–2016.

| Year | Moran's I | Z-Value | *p*-Value |
|------|-----------|---------|-----------|
| 2007 | 0.1673 | 1.9845 | 0.0290 |
| 2008 | 0.1878 | 2.1384 | 0.0200 |
| 2009 | 0.2222 | 2.566 | 0.0080 |
| 2010 | 0.1881 | 2.2663 | 0.0120 |
| 2011 | 0.2206 | 2.3556 | 0.0170 |
| 2012 | 0.2177 | 0.3799 | 0.0160 |
| 2013 | 0.2067 | 2.3203 | 0.0120 |
| 2014 | 0.2062 | 2.2560 | 0.0140 |
| 2015 | 0.2040 | 2.3803 | 0.0130 |
| 2016 | 0.2468 | 2.6108 | 0.0050 |

Moran scatter plots for 2007 and 2016 were drawn through GeoDa software, as shown in Figures 7 and 8 and Table 6. In 2007, ten provinces (Tianjing, Jilin, Heilongjiang, Shanghai, Jiangsu, Zhejiang, Anhui, Fujian, Shandong, and Guangdong) belonged to HH clustering patterns. This indicates that the eastern region boasted a high coordination degree, small differences, and close spatial links. Two provinces (Hebei and Jiangxi) belonged to HH clustering patterns, and a total of twelve provinces belonged to the LH clustering pattern. In addition, Beijing, Hunan, Chongqing, Shaanxi, and Xinjiang belonged to the HL clustering pattern, with their coordination levels higher than those of surrounding areas. After ten years of development, in 2016, the number of provinces under the HH clustering pattern increased to 11, among which Beijing, Hubei, and Hunan were newly included, while Jilin and Heilongjiang were newly excluded. The number of provinces under the LH clustering pattern increased to three, among which Henan was newly included. The number of provinces under the LL clustering pattern remained at 12, but the provinces changed. For example, Shaanxi and Xinjiang changed from HL to LL. Only Heilongjiang and Chongqing belonged to the HL clustering pattern. It can be seen that the spatial clustering pattern of most provinces and their adjacent areas remained unchanged and had certain spatial stability.

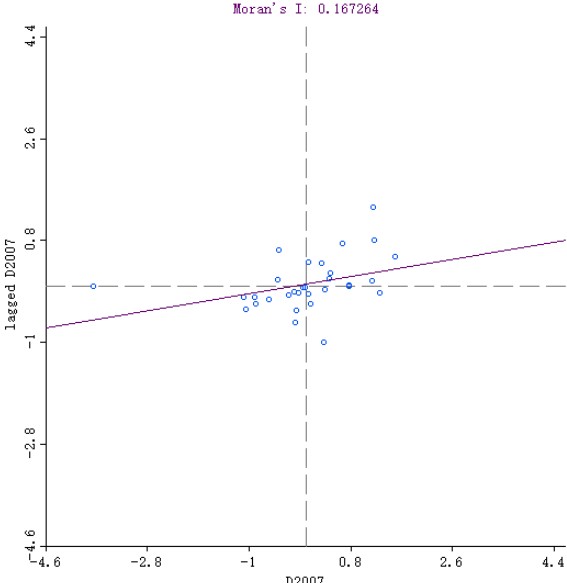

**Figure 7.** Moran scatter of coordination between GF and GE in China in 2007.

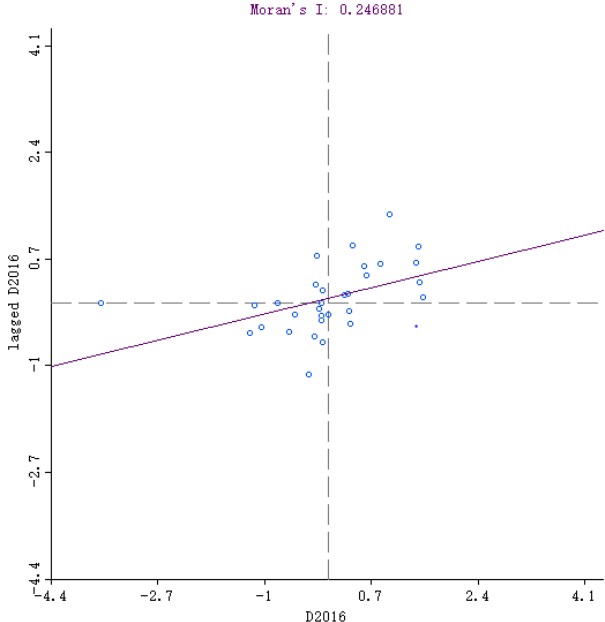

**Figure 8.** Moran scatter of coordination between GF and GE in China in 2016.

**Table 6.** Results of Moran scatter chart.

| Province | 2007 | 2016 | Province | 2007 | 2016 |
|---|---|---|---|---|---|
| Beijing | HL | HH | Henan | LL | LH |
| Tianjing | HH | HH | Hubei | LL | HH |
| Hebei | LH | LH | Hunan | HL | HH |
| Shanxi | LL | LL | Guangdong | HH | HH |
| Inner Mongolia | LL | LL | Guangxi | LL | / |
| Liaoning | LL | LL | Hainan | / | / |
| Jilin | HH | LL | Chongqing | HL | HL |
| Heilongjiang | HH | HL | Sichuan | LL | LL |
| Shanghai | HH | HH | Guizhou | LL | LL |
| Jiangsu | HH | HH | Yunnan | LL | LL |
| Zhejiang | HH | HH | Shaanxi | HL | LL |
| Anhui | HH | HH | Gansu | LL | LL |
| Fujian | HH | HH | Qinghai | LL | LL |
| Jiangxi | LH | LH | Ningxia | LL | LL |
| Shandong | HH | HH | Xinjiang | HL | LL |

### 4.4. Analysis and Forecast of Dynamic Spatial and Temporal Evolutions of Coordination

In this paper, the space Markov chain calculation code in PySAL was used to calculate the spatial Markov transfer probability matrix P (Table 7) and traversal probability distribution matrix Π (Table 8). Specifically, (1) from Table 7, when the observation value was the same as its space lag, the probability that the observed value maintained the current state was the largest, compared with other cases. When spatial lag was not considered, the probability that the observed value maintains its current state was not always the maximum. For example, when the spatial lag and the observed value were at S4 and S3, respectively, the probability that the observed value kept S3 was 0.4 while the probability that it would rise to S4 was 0.6. This demonstrates that the coordination value of GF and the GE in China was not always stable and was affected by its surrounding areas. (2) When the observed value and the spatial lag were in the lowest state S1, the sum of probability that the observed value goes upward from the current state was 0.2857. With the increase of spatial lag, the probabilities that the observed value goes upward were 0.1875, 0.1429, 0.2667, and 0.5, exhibiting a trend of first decreasing and then rising. In short, if the coordinated development of a province was poor while those of its

surrounding provinces were excellent, the case is conducive to the advancement of its coordination value. (3) When the observed value and the space lag were in the highest state S5, with the decrease of space lag, the probability that the observed value maintained S5 was barely influenced by space lag and always remained 1, except the case where the space lag was S1 (the coordinated development state of a province is S5 while that of surrounding provinces was S1, which rarely occurs). This showed that when a province boasted an excellent coordinated development, the state of its neighboring provinces had a limited influence on it and did not affect its main development trend. (4) According to the traversal probability distribution matrix (Table 8), the probability that the coordinated development state of an observed province and its surrounding provinces stays in the same level is the largest, which also suggests that the coordinated development level of GF and the GE showed the HH and LL clustering patterns in the long run.

**Table 7.** One-step space Markov transfer probability matrix for coordination between GF and the GE.

| Spatial Lag | Observed Value | *t/t + 1* | | | | |
| --- | --- | --- | --- | --- | --- | --- |
| | | S1 | S2 | S3 | S4 | S5 |
| S1 | S1 | 0.7143 | 0.2857 | 0.0000 | 0.0000 | 0.0000 |
| | S2 | 0.2000 | 0.4000 | 0.4000 | 0.0000 | 0.0000 |
| | S3 | 0.0000 | 0.0000 | 0.7143 | 0.2857 | 0.0000 |
| | S4 | 0.0000 | 0.0000 | 0.0625 | 0.5625 | 0.3750 |
| | S5 | 0.0000 | 0.0000 | 0.0000 | 0.2500 | 0.7500 |
| S2 | S1 | 0.8125 | 0.1875 | 0.0000 | 0.0000 | 0.0000 |
| | S2 | 0.0000 | 0.5000 | 0.5000 | 0.0000 | 0.0000 |
| | S3 | 0.0000 | 0.0000 | 0.7143 | 0.2857 | 0.0000 |
| | S4 | 0.0000 | 0.0000 | 0.0000 | 0.8000 | 0.2000 |
| | S5 | 0.0000 | 0.0000 | 0.0000 | 0.0000 | 1.0000 |
| S3 | S1 | 0.8571 | 0.1429 | 0.0000 | 0.0000 | 0.0000 |
| | S2 | 0.0000 | 0.6000 | 0.4000 | 0.0000 | 0.0000 |
| | S3 | 0.0000 | 0.0000 | 0.6000 | 0.4000 | 0.0000 |
| | S4 | 0.0000 | 0.0000 | 0.0000 | 0.5714 | 0.4286 |
| | S5 | 0.0000 | 0.0000 | 0.0000 | 0.0000 | 1.0000 |
| S4 | S1 | 0.7333 | 0.2667 | 0.0000 | 0.0000 | 0.0000 |
| | S2 | 0.0000 | 0.8000 | 0.2000 | 0.0000 | 0.0000 |
| | S3 | 0.0000 | 0.0000 | 0.4000 | 0.6000 | 0.0000 |
| | S4 | 0.0000 | 0.0000 | 0.1429 | 0.7143 | 0.1429 |
| | S5 | 0.0000 | 0.0000 | 0.0000 | 0.0000 | 1.0000 |
| S5 | S1 | 0.5000 | 0.5000 | 0.0000 | 0.0000 | 0.0000 |
| | S2 | 0.0476 | 0.7143 | 0.2381 | 0.0000 | 0.0000 |
| | S3 | 0.0000 | 0.0000 | 0.6667 | 0.3333 | 0.0000 |
| | S4 | 0.0000 | 0.0000 | 0.0000 | 0.9000 | 0.1000 |
| | S5 | 0.0000 | 0.0000 | 0.0000 | 0.0000 | 1.0000 |

**Table 8.** One-step spatial Markov traversal distribution probability of coordination between GF and the GE.

| Spatial Lag / Observed Value | S1 | S2 | S3 | S4 | S5 |
| --- | --- | --- | --- | --- | --- |
| S1 | 0.7451 | 0.2549 | 0.0000 | 0.0000 | 0.0000 |
| S2 | 0.0385 | 0.6538 | 0.3077 | 0.0000 | 0.0000 |
| S3 | 0.0000 | 0.0000 | 0.6531 | 0.3469 | 0.0000 |
| S4 | 0.0000 | 0.0000 | 0.0444 | 0.6889 | 0.2667 |
| S5 | 0.0000 | 0.0000 | 0.0000 | 0.0465 | 0.9535 |

The LISA Markov chain was also calculated by using the PySAL module in Python software. The results are given in Table 9.

**Table 9.** LISA Markov transfer probability matrix.

| Clustering Pattern | HH | LH | LL | HL |
|---|---|---|---|---|
| HH | 0.9643 | 0.0000 | 0.0000 | 0.0357 |
| LH | 0.0588 | 0.8824 | 0.0588 | 0.0000 |
| LL | 0.0000 | 0.0268 | 0.8929 | 0.0804 |
| HL | 0.0364 | 0.0000 | 0.1818 | 0.7818 |

According to Table 9, the following conclusions can be drawn:

(1) The probabilities on the principal diagonal of the LISA Markov one-step transfer probability matrix, which all exceeded 0.7, were much larger than the other probabilities. This indicates that the clustering pattern of coordination values between GF and the GE in 30 Chinese provinces remained stable, especially the HH clustering pattern that reached 0.9643.

(2) From the perspective of outward diffusion path, the probability that the clustering pattern of the coordination values between GF and the GE in 30 Chinese provinces transfers from HL to HH was 0.0364, while the probability that it transfers from HL to LH was 0. That is, the probability for outward diffusion of the clustering pattern of coordination values between GF and GE in 30 Chinese provinces was 0.0364.

(3) From the perspective of the inward diffusion path, the probability that the clustering pattern of coordination values between urbanization and ecological environment of 30 Chinese provinces transferred from LH to HH was 0.0588, while the probability that it transferred from LH to HL was 0. That is, the probability for inward diffusion of the clustering pattern of coordination values between urbanization and ecological environment of 30 Chinese provinces was 0.0588.

(4) According to the classification of saturation type and substitution type, the transfer probability of saturation type was 0 while that of substitution type was 0.0952. The coordination between urbanization and ecological environment in 30 Chinese provinces did not experience saturation change, indicating that the coordination value was affected by the surrounding areas to some extent, but did not change much.

## 5. Discussion and Conclusions

This paper aimed at investigating the coupling coordination between the Chinese GF and GE systems. Since GF and the GE are coupled systematically and complexly, it was crucial to employ the coupling coordination degree model in order to grasp the cooperative interaction and feedback among different determinants. Besides, for overcoming subjectivity or computational complexity, the study determined the weights of indicators in the GE system by means of an entropy weighting method. This study is of practical significance for investigation and research on the coordinated development of GF and the GE.

The Chinese GF and GE share a complex relationship, and their coupling coordination degree exhibits an upward trend. However, the Eastern, Central, Western and Northeastern regions still differ in terms of coordination states. To be specific, the Eastern region boasts the highest level of coordinated development of GF and the GE, reaching 0.6 in 2014 and realizing the transformation from barely balanced development to favorably balanced development. At present, the coordination values of GF and the GE in the Central and Northeastern regions lie between 0.5 and 0.6, and they are still in the phase of barely balanced development. The coordinated development of GF and GE in the western region is relatively backward and remains in the phase of slightly unbalanced development.

Compared with previous studies, it is innovative and enlightening to analyze the spatial correlation of coupling coordination between various regions. Researchers have probed into the evolving distribution of high-pollution industries [50,51]. In fact, for the sake of local green development,

the local governments of China's developed areas would rather transfer high-pollution industries to developing areas than reduce pollution. This phenomenon can explain the evolving distribution. In terms of global correlation, the differences of coordination between GF and the GE among the provinces in China are not randomly distributed in space, but positively correlated, showing a strong spatial dependence on the whole. From the perspective of local correlation, most provinces are in HH and LL clustering patterns, and the spatial clustering patterns of most provinces and their adjacent provinces experience no transfer and remain spatially stable.

The spatial clustering pattern evolution of the coordination values of GF and the GE were analyzed on the basis of the LISA Markov chain. The results reveal that the clustering pattern of the coordination values of the GF and GE of 30 Chinese provinces remain stable, especially the HH clustering pattern which reached 0.9643.

Some developed countries are more mature in green finance and green economy development and may provide some experience to China. For example, the United States, as a developed economy, had an early start in green finance. Since the 1970s, the U.S. Congress has passed more than 20 laws on environmental protection related to water environment, air pollution, waste management, and the cleanup of polluted sites, etc. In 1980, the U.S. federal government introduced the Comprehensive Environmental Response, Compensation and Liability Act, which makes banks responsible for environmental pollution caused by their customers. In addition, the European Union (EU) attaches great importance to the development of green finance, with a mature legal system and active product innovation. The EU incentivizes green projects through tax incentives and government guarantees. For example: the German government gives certain subsidies and interest rates for loans for green projects; the European Union provides tax incentives for green credits and securitized products; and the British government uses a "loan guarantee scheme" to support small and medium-sized enterprises (SMEs), especially environmental SMEs.

In recent years, green finance has developed rapidly in countries with an emerging economy. The Central Bank of Brazil introduced a new regulatory approach in April 2014 that required commercial banks to develop strategic actions and governance frameworks for environmental and social risk management and to implement them as core elements of overall risk management. Currently about 10% of bank loans in Brazil are classified as green loans. In particular, the central bank of Bangladesh has made increasing financial inclusion an explicit objective of monetary policy and has provided credit guidelines for commercial banks that include new energy, pollution control, and energy efficiency, etc. The share of green credit now stands at 5%.

Compared to these countries discussed above, there are a number of problems and challenges in green finance development for China, such as: (a) the overall proportion of investment and loans for green projects by financial institutions is still low; (b) the lack of a green financial indicator system and incentive mechanisms; (c) the government's policy of supporting green industries is inadequate; (d) the failure to establish a virtuous circular market mechanism for building ecological civilization. Therefore, it is necessary for China to learn from the experience of some developed countries and further improve the construction of green finance laws and regulations, insist on market-based operation, and encourage market innovation as a long-term mechanism for optimizing the allocation of green finance funds, enhancing the efficiency of the use of green finance funds and building China's green finance development.

This paper may contribute to the realization of social and economic sustainable development through exploring the coronation between green finance and the green economy. Our results prove that there is a certain gap in the coordinated development of green finance and the green economy for 30 provinces in China. Therefore, in order to further narrow this gap, it is necessary to strengthen the construction of a diversified green financial system in the central and western regions, for example, by vigorously develop green bonds, energy conservation and environmental protection risk investment funds, and introducing clean development trading projects, as well as avoiding the transfer of polluting industries to the central and western regions, especially in areas with weak environmental and

ecological carrying capacity, so as to achieve comprehensive coordination between regions, promote green finance and the green economy, and achieve the country's overall social sustainable development.

In addition, this paper also provides some suggestions for the government about how to effectively execute financial policies. Specifically, policy-makers need to recognize the coupling and coordination between green finance and the green economy. It is important not to focus only on changes in the scale of one, but to make it a policy objective to promote coordinated development between the two. The first suggestion is to improve the investment channels of the green financial system and raise the level of investment; the second is to improve the relevant policies and provide policy support for the development of green industries in order to promote the balanced development of the two systems. Also, the regulator should establish a monitoring and feedback system between the two systems and enhance the operational efficiency of the coupled system through a positive feedback effect. It is important to establish and improve a system for monitoring the use of funds by enterprises, earmarking funds for green enterprises and improving the efficiency of their use. A feedback mechanism should also be improved to fully receive feedback on the effectiveness of the use of green finance policies. Policies already enacted should be revised based on market feedback to better promote green industries. At the same time, green financial institutions should be guided to establish information service platforms, to strengthen exchanges with green enterprises, and to better understand the financing needs of energy-saving and environmental protection enterprises, and to provide them with matching financing solutions.

Our application of the spatial Markov approach to the field of environmental finance, in particular to the coordinated development of GF and the GE, provides new ideas and insights for enriching the research literature on green finance and the green economy and coordinated development. Therefore, in order to better achieve the coordinated development of green finance and the green economy, we will further explore the path of green finance for the green economy and the dynamic coordination between the two.

However, this study has certain limitations. First, in terms of the construction of green finance indicators, generally, the green financial system can be divided into five areas, namely, green credit, green securities, green insurance, green investment, and carbon finance. But this paper does not include carbon finance in the indicator system due to a serious lack of data, even if only a small percentage of the development of green finance is in carbon finance. Second, this paper evaluates the spatial distribution difference and dynamic evolution trend of the coordination by introducing global/local spatial autocorrelation, a space Markov chain, and a local indicators of spatial association Markov chain. However, considering the accuracy of results, we only predict coordination in the short term, but not in the long term (e.g., over the next 10–20 years). In the future, we hope to take a more appropriate approach to predicting the long-term degree of coordination. Third, this paper predicts the degree of coordination between GF and the GE, but there is no further research on the early warning model of coordination. Therefore, in the future we will explore the early warning model of coordinated development of GF and the GE to provide better policy recommendations for sustainable development.

**Author Contributions:** Data curation, Y.R. and J.L.; Formal analysis, N.L.; Funding acquisition, N.L. and C.L.; Investigation, N.L. and Y.X.; Methodology, N.L. and Y.X.; Resources, Y.X. and J.L.; Software, N.L. and Y.R.; Supervision, C.L.; Validation, Y.R.; Visualization, J.L.; Writing—original draft, N.L.; Writing—review & editing, C.L. All authors have read and agreed to the published version of the manuscript.

**Funding:** This research was funded by [Outstanding Innovation Scholarship for Doctoral Candidate of CUMT] grant number [No.2019YCBS034].

**Conflicts of Interest:** The authors declare no conflict of interest.

## Appendix A

**Table A1.** The data on green finance in 2016.

| Province | X1 | X2 | X3 | X4 | X5 | X6 | X7 |
|---|---|---|---|---|---|---|---|
| Anhui | 31.533 | 20.398 | 21.465 | 13.603 | 118.900 | 2.420 | 2.041 |
| Beijing | 34.071 | 17.629 | 6.845 | 2.267 | 94.768 | 5.672 | 2.627 |
| Chongqing | 41.945 | 34.607 | 11.940 | 2.013 | 69.658 | 3.403 | 0.813 |
| Fujian | 45.834 | 17.482 | 10.248 | 3.434 | 139.718 | 2.522 | 0.658 |
| Gansu | 78.430 | 15.196 | 26.338 | 12.067 | 81.317 | 3.024 | 1.633 |
| Guangdong | 34.979 | 21.542 | 5.263 | 1.157 | 40.302 | 2.212 | 0.455 |
| Guangxi | 54.675 | 31.085 | 18.038 | 6.285 | 59.575 | 2.014 | 1.115 |
| Guizhou | 89.387 | 0.893 | 3.016 | 3.197 | 41.365 | 2.982 | 1.005 |
| Hainan | 88.502 | 9.152 | 5.218 | 13.796 | 103.034 | 2.513 | 0.748 |
| Hebei | 54.816 | 31.337 | 21.232 | 5.618 | 56.942 | 4.344 | 1.246 |
| Heilongjiang | 45.927 | 31.839 | 5.910 | 38.563 | 118.039 | 2.684 | 1.128 |
| Henan | 44.654 | 24.928 | 20.418 | 7.948 | 53.693 | 2.626 | 0.889 |
| Hubei | 47.261 | 45.737 | 11.276 | 3.716 | 102.804 | 2.267 | 1.423 |
| Hunan | 43.103 | 34.536 | 10.931 | 9.950 | 58.951 | 2.695 | 0.635 |
| Inner Mongolia | 69.363 | 25.146 | 46.952 | 21.000 | 55.273 | 3.532 | 2.515 |
| Jiangsu | 35.388 | 28.381 | 11.267 | 2.750 | 73.939 | 2.856 | 0.989 |
| Jiangxi | 49.799 | 15.179 | 35.197 | 5.436 | 67.965 | 2.537 | 1.694 |
| Jilin | 46.975 | 11.009 | 12.799 | 13.093 | 67.719 | 3.406 | 0.569 |
| Liaoning | 54.347 | 34.251 | 24.137 | 3.325 | 53.353 | 1.906 | 0.792 |
| Ningxia | 68.581 | 49.764 | 32.346 | 14.361 | 96.470 | 2.950 | 3.194 |
| Qinghai | 90.605 | 25.812 | 43.819 | 16.800 | 100.128 | 4.814 | 2.189 |
| Shaanxi | 52.228 | 10.505 | 8.578 | 3.332 | 50.176 | 2.889 | 1.636 |
| Shandong | 37.252 | 24.067 | 20.249 | 2.955 | 47.212 | 2.733 | 1.148 |
| Shanghai | 33.165 | 17.918 | 6.574 | 1.242 | 45.535 | 1.943 | 0.729 |
| Shanxi | 34.000 | 4.888 | 21.987 | 3.369 | 59.801 | 3.370 | 4.028 |
| Sichuan | 55.777 | 27.700 | 20.928 | 7.411 | 53.879 | 2.077 | 0.882 |
| Tianjing | 22.050 | 26.143 | 0.687 | 2.079 | 68.171 | 1.774 | 0.299 |
| Xinjiang | 67.222 | 29.210 | 12.244 | 28.760 | 75.765 | 1.572 | 3.242 |
| Yunnan | 64.132 | 17.220 | 28.497 | 6.377 | 67.249 | 2.991 | 0.986 |
| Zhejiang | 30.266 | 24.704 | 10.075 | 1.364 | 69.104 | 2.314 | 1.377 |

## Appendix B

**Table A2.** The data on green economy in 2016.

| Province | D11 | D12 | D13 | D21 | D22 | D23 | D24 | P11 | P12 | P13 | P14 |
|---|---|---|---|---|---|---|---|---|---|---|---|
| Anhui | 39561 | 15466 | 10.9 | 180.2 | 11.9 | 45.5 | 21.8 | 2.0 | 73342 | 14211 | 400388 |
| Beijing | 118198 | 48883 | 11.5 | 173.1 | 24.3 | 53.9 | 7.6 | 3.2 | 8515 | 629 | 10257 |
| Chongqing | 58502 | 21032 | 12.9 | 151.6 | 10.7 | 62.6 | 12.2 | 3.0 | 36089 | 2607 | 383649 |
| Fujian | 74707 | 23355 | 10.9 | 191.5 | 15.3 | 45.1 | 14.4 | 3.2 | 80684 | 5080 | 297747 |
| Gansu | 27643 | 13086 | 6.0 | 125.3 | 9.2 | 51.5 | 15.4 | 1.6 | 17827 | 5523 | 457213 |
| Guangdong | 74016 | 28495 | 11.0 | 246.1 | 14.2 | 42.3 | 13.1 | 2.8 | 146816 | 5554 | 602485 |
| Guangxi | 38027 | 15013 | 9.0 | 256.4 | 9.8 | 46.4 | 17.1 | 2.1 | 54856 | 6056 | 344756 |
| Guizhou | 33246 | 14666 | 12.1 | 172.0 | 11.4 | 59.2 | 12.1 | 2.9 | 26049 | 6732 | 510624 |
| Hainan | 44347 | 18431 | 9.5 | 253.1 | 11.3 | 44.0 | 17.8 | 2.2 | 5948 | 346 | 31512 |
| Hebei | 43062 | 14328 | 7.6 | 132.0 | 13.7 | 48.3 | 18.9 | 4.0 | 81582 | 33236 | 656826 |
| Heilongjiang | 40432 | 17393 | 2.0 | 117.4 | 13.6 | 57.9 | 13.7 | 3.2 | 31576 | 8900 | 248653 |
| Henan | 42575 | 16043 | 9.4 | 115.6 | 10.9 | 54.7 | 13.0 | 2.4 | 131594 | 13617 | 811530 |
| Hubei | 55665 | 19391 | 10.5 | 204.3 | 12.8 | 61.3 | 16.1 | 2.9 | 79986 | 7502 | 437665 |
| Hunan | 46382 | 17490 | 9.2 | 217.1 | 15.1 | 62.4 | 14.6 | 2.3 | 71857 | 7323 | 475759 |
| Inner Mongolia | 72064 | 22293 | 1.7 | 103.4 | 10.3 | 55.3 | 23.5 | 7.7 | 32505 | 24762 | 964549 |
| Jiangsu | 96887 | 35875 | 10.4 | 215.4 | 16.6 | 55.4 | 25.4 | 3.9 | 207976 | 10482 | 725691 |
| Jiangxi | 40400 | 16040 | 10.6 | 171.4 | 8.9 | 45.5 | 17.3 | 1.9 | 90027 | 12665 | 262705 |
| Jilin | 53868 | 13786 | 5.1 | 124.5 | 10.3 | 55.3 | 15.0 | 2.9 | 35629 | 5865 | 285486 |

**Table A2.** *Cont.*

| Province | D11 | D12 | D13 | D21 | D22 | D23 | D24 | P11 | P12 | P13 | P14 |
|---|---|---|---|---|---|---|---|---|---|---|---|
| Liaoning | 50791 | 23670 | -22.4 | 146.3 | 12.9 | 65.0 | 13.0 | 4.8 | 76268 | 22822 | 816094 |
| Ningxia | 47194 | 18570 | 8.8 | 187.8 | 13.5 | 53.8 | 21.8 | 24.2 | 17850 | 3185 | 270674 |
| Qinghai | 43531 | 16751 | 6.4 | 170.3 | 14.5 | 58.6 | 11.0 | 9.4 | 8891 | 17794 | 114708 |
| Shaanxi | 51015 | 16657 | 7.6 | 159.3 | 16.0 | 59.1 | 15.4 | 1.9 | 39365 | 10026 | 534787 |
| Shandong | 68733 | 25860 | 8.0 | 132.8 | 15.9 | 54.4 | 24.7 | 3.9 | 193087 | 20416 | 1096994 |
| Shanghai | 116562 | 49617 | 12.2 | 200.9 | 12.7 | 53.4 | 4.4 | 4.8 | 50144 | 1813 | 70764 |
| Shanxi | 35532 | 15065 | 2.2 | 114.5 | 9.4 | 51.5 | 14.8 | 5.3 | 34727 | 28845 | 752676 |
| Sichuan | 40003 | 16013 | 9.6 | 214.6 | 12.9 | 62.8 | 13.7 | 2.5 | 75962 | 10647 | 533853 |
| Tianjing | 115053 | 36257 | 8.1 | 114.0 | 18.1 | 42.1 | 15.4 | 5.3 | 18022 | 1490 | 56701 |
| Xinjiang | 40564 | 15247 | 3.5 | 167.2 | 15.2 | 65.4 | 19.8 | 181.7 | 24594 | 6772 | 538996 |
| Yunnan | 31093 | 14534 | 8.6 | 132.0 | 13.2 | 53.1 | 15.8 | 2.2 | 52168 | 13747 | 470906 |
| Zhejiang | 84916 | 30743 | 10.2 | 187.2 | 16.3 | 51.9 | 17.7 | 3.6 | 145354 | 4431 | 490191 |

Continued

| Province | S11 | S12 | S13 | S14 | R11 | R12 | R13 | R21 | R22 | R23 |
|---|---|---|---|---|---|---|---|---|---|---|
| Anhui | 14.02 | 15.91 | 41.71 | 27.53 | 4.70 | 41.05 | 60983 | 507.7 | 92.80 | 99.94 |
| Beijing | 16.01 | 37.79 | 48.40 | 35.84 | 4.46 | 80.23 | 100578 | 630.9 | 86.33 | 99.84 |
| Chongqing | 16.86 | 19.61 | 40.76 | 38.43 | 1.29 | 48.13 | 42738 | 289.7 | 84.30 | 99.98 |
| Fujian | 13.08 | 17.36 | 43.32 | 65.95 | 0.00 | 42.88 | 67142 | 382.8 | 65.10 | 98.44 |
| Gansu | 13.94 | 10.09 | 31.50 | 11.28 | 0.83 | 51.41 | 7975 | 131.7 | 51.81 | 72.76 |
| Guangdong | 17.87 | 41.16 | 42.39 | 51.26 | 5.53 | 52.01 | 259032 | 2039.1 | 95.86 | 96.22 |
| Guangxi | 11.77 | 17.46 | 37.62 | 56.51 | 1.04 | 39.56 | 14858 | 718.0 | 55.86 | 98.96 |
| Guizhou | 14.98 | 11.48 | 36.80 | 37.09 | 1.63 | 44.67 | 10425 | 183.9 | 61.98 | 94.65 |
| Hainan | 12.02 | 16.65 | 40.30 | 55.38 | 1.14 | 54.25 | 1939 | 90.9 | 75.94 | 99.94 |
| Hebei | 14.31 | 11.44 | 40.80 | 23.41 | 1.21 | 41.54 | 31826 | 607.0 | 73.27 | 97.80 |
| Heilongjiang | 11.91 | 20.28 | 35.35 | 43.16 | 1.06 | 54.04 | 18046 | 758.2 | 50.45 | 80.62 |
| Henan | 10.43 | 10.01 | 39.33 | 21.50 | 1.29 | 41.78 | 49145 | 679.7 | 78.21 | 98.75 |
| Hubei | 10.99 | 13.97 | 37.60 | 38.40 | 2.96 | 43.94 | 41822 | 687.7 | 59.26 | 95.80 |
| Hunan | 10.57 | 9.01 | 40.60 | 47.77 | 1.13 | 46.37 | 34050 | 613.1 | 67.78 | 99.89 |
| Inner Mongolia | 19.77 | 26.01 | 39.85 | 21.03 | 0.72 | 43.78 | 5846 | 245.5 | 37.32 | 98.87 |
| Jiangsu | 14.79 | 35.24 | 42.94 | 15.80 | 3.82 | 50.00 | 231033 | 1742.9 | 94.80 | 99.93 |
| Jiangxi | 14.16 | 12.36 | 43.63 | 60.01 | 1.81 | 41.97 | 31472 | 259.9 | 57.50 | 94.97 |
| Jilin | 13.37 | 17.05 | 34.97 | 40.38 | 1.14 | 42.45 | 9995 | 318.7 | 48.42 | 86.30 |
| Liaoning | 11.33 | 26.63 | 36.35 | 38.24 | 1.35 | 51.55 | 25104 | 831.4 | 25.40 | 93.27 |
| Ningxia | 18.30 | 37.17 | 40.43 | 11.89 | 1.40 | 45.40 | 2677 | 91.0 | 49.86 | 98.28 |
| Qinghai | 10.78 | 10.37 | 31.12 | 5.63 | 0.71 | 42.81 | 1357 | 51.8 | 42.20 | 96.28 |
| Shaanxi | 12.30 | 15.39 | 40.14 | 41.42 | 1.41 | 42.35 | 48455 | 348.0 | 67.79 | 98.53 |
| Shandong | 17.91 | 22.70 | 42.26 | 16.73 | 1.91 | 46.68 | 98093 | 1069.9 | 88.41 | 100.00 |
| Shanghai | 7.83 | 53.25 | 38.60 | 10.74 | 4.94 | 69.78 | 64230 | 806.9 | 94.72 | 100.00 |
| Shanxi | 11.86 | 11.67 | 40.52 | 18.03 | 1.01 | 55.45 | 10062 | 257.9 | 46.98 | 94.60 |
| Sichuan | 12.47 | 12.17 | 39.90 | 35.22 | 1.26 | 47.23 | 62445 | 609.5 | 45.85 | 98.60 |
| Tianjing | 10.59 | 21.38 | 37.22 | 9.87 | 3.38 | 56.44 | 39734 | 291.5 | 97.81 | 94.16 |
| Xinjiang | 12.22 | 27.00 | 38.51 | 4.24 | 1.09 | 45.12 | 7116 | 255.3 | 58.23 | 83.30 |
| Yunnan | 11.33 | 9.03 | 37.84 | 50.03 | 0.93 | 46.68 | 12032 | 250.9 | 51.93 | 92.96 |
| Zhejiang | 13.17 | 27.61 | 41.02 | 59.07 | 3.86 | 50.99 | 221456 | 1001.8 | 95.20 | 99.98 |

## Appendix C

**Table A3.** The coordination between green finance and the green economy.

| Province | 2007 | 2008 | 2009 | 2010 | 2011 | 2012 | 2013 | 2014 | 2015 | 2016 |
|---|---|---|---|---|---|---|---|---|---|---|
| Anhui | 0.40 | 0.41 | 0.43 | 0.46 | 0.50 | 0.51 | 0.53 | 0.54 | 0.56 | 0.60 |
| Beijing | 0.53 | 0.52 | 0.51 | 0.52 | 0.58 | 0.59 | 0.62 | 0.66 | 0.72 | 0.72 |
| Chongqing | 0.40 | 0.43 | 0.43 | 0.46 | 0.51 | 0.51 | 0.53 | 0.53 | 0.54 | 0.56 |
| Fujian | 0.46 | 0.47 | 0.48 | 0.49 | 0.52 | 0.52 | 0.54 | 0.54 | 0.56 | 0.57 |
| Gansu | 0.28 | 0.29 | 0.27 | 0.29 | 0.29 | 0.30 | 0.33 | 0.33 | 0.34 | 0.36 |
| Guangdong | 0.51 | 0.55 | 0.57 | 0.62 | 0.63 | 0.64 | 0.66 | 0.67 | 0.71 | 0.73 |
| Guangxi | 0.38 | 0.39 | 0.41 | 0.43 | 0.45 | 0.46 | 0.47 | 0.48 | 0.50 | 0.50 |
| Guizhou | 0.30 | 0.33 | 0.34 | 0.35 | 0.37 | 0.27 | 0.38 | 0.39 | 0.40 | 0.35 |
| Hainan | 0.47 | 0.47 | 0.47 | 0.50 | 0.55 | 0.52 | 0.49 | 0.56 | 0.50 | 0.40 |
| Hebei | 0.34 | 0.36 | 0.39 | 0.40 | 0.42 | 0.41 | 0.43 | 0.44 | 0.48 | 0.49 |
| Heilongjiang | 0.44 | 0.45 | 0.45 | 0.46 | 0.50 | 0.51 | 0.56 | 0.55 | 0.55 | 0.57 |
| Henan | 0.37 | 0.39 | 0.40 | 0.42 | 0.43 | 0.42 | 0.44 | 0.46 | 0.48 | 0.50 |
| Hubei | 0.39 | 0.42 | 0.43 | 0.45 | 0.48 | 0.48 | 0.52 | 0.53 | 0.54 | 0.56 |
| Hunan | 0.43 | 0.43 | 0.46 | 0.48 | 0.50 | 0.51 | 0.51 | 0.52 | 0.54 | 0.55 |
| Inner Mongolia | 0.36 | 0.35 | 0.39 | 0.41 | 0.42 | 0.41 | 0.43 | 0.43 | 0.43 | 0.44 |
| Jiangsu | 0.52 | 0.54 | 0.58 | 0.61 | 0.67 | 0.68 | 0.69 | 0.70 | 0.72 | 0.72 |
| Jiangxi | 0.34 | 0.39 | 0.40 | 0.39 | 0.42 | 0.40 | 0.43 | 0.43 | 0.47 | 0.49 |
| Jilin | 0.47 | 0.44 | 0.47 | 0.46 | 0.47 | 0.47 | 0.48 | 0.48 | 0.49 | 0.49 |
| Liaoning | 0.39 | 0.41 | 0.42 | 0.45 | 0.47 | 0.48 | 0.49 | 0.51 | 0.52 | 0.50 |
| Ningxia | 0.37 | 0.40 | 0.39 | 0.41 | 0.43 | 0.42 | 0.44 | 0.46 | 0.50 | 0.49 |
| Qinghai | 0.30 | 0.25 | 0.28 | 0.29 | 0.32 | 0.30 | 0.32 | 0.31 | 0.33 | 0.34 |
| Shaanxi | 0.40 | 0.42 | 0.41 | 0.43 | 0.45 | 0.46 | 0.48 | 0.49 | 0.52 | 0.52 |
| Shandong | 0.44 | 0.48 | 0.49 | 0.52 | 0.55 | 0.56 | 0.58 | 0.57 | 0.59 | 0.60 |
| Shanghai | 0.52 | 0.51 | 0.56 | 0.56 | 0.62 | 0.62 | 0.62 | 0.65 | 0.65 | 0.65 |
| Shanxi | 0.32 | 0.35 | 0.37 | 0.41 | 0.41 | 0.42 | 0.46 | 0.46 | 0.48 | 0.50 |
| Sichuan | 0.37 | 0.39 | 0.42 | 0.44 | 0.47 | 0.47 | 0.49 | 0.47 | 0.50 | 0.50 |
| Tianjing | 0.42 | 0.44 | 0.44 | 0.46 | 0.52 | 0.50 | 0.53 | 0.53 | 0.54 | 0.63 |
| Xinjiang | 0.43 | 0.42 | 0.42 | 0.44 | 0.47 | 0.46 | 0.48 | 0.49 | 0.47 | 0.47 |
| Yunnan | 0.28 | 0.32 | 0.32 | 0.32 | 0.36 | 0.34 | 0.36 | 0.36 | 0.36 | 0.43 |
| Zhejiang | 0.56 | 0.58 | 0.60 | 0.62 | 0.66 | 0.67 | 0.68 | 0.69 | 0.71 | 0.71 |

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
