# Peer review of "Examining the Coordination Between Green Finance and Green Economy Aiming for Sustainable Development: A Case Study of China"

_sustainability, doi:10.3390/su12093717_

Round 1

Reviewer 1 Report

Dear Authors, 

I think the topic is very interesting and the problem is current.  The scientific foundation of the topic and the application of the method are at a high level.

Should be changed to the following:

  • the legend is barely visible on the maps,
  • the basic data of the studies should be made public so that the analysis can be performed / verified by anyone. It would be necessary to indicate the primary data in an annex, in a summary table.

Author Response

Response to Reviewer

Nana Liua, Chuanzhe Liua*, Yufei Xiab, Yi Rena , Jinzhi Lianga

At first, we appreciate editors’ and reviewers’ valuable comments and suggestions which help us improve the paper significantly.

Response to Reviewer 1 Comments

Point 1: Dear Authors, I think the topic is very interesting and the problem is current.  The scientific foundation of the topic and the application of the method are at a high level.

Should be changed to the following: the legend is barely visible on the maps,

Response1: we are very grateful to you for your comments. We have modified the graphics to make the legend as visual as possible.

Point 2: the basic data of the studies should be made public so that the analysis can be performed / verified by anyone. It would be necessary to indicate the primary data in an annex, in a summary table.

Response2: Thanks for your suggestion. We have added primary data on green finance and the green economy and the value of coordination in the appendix.

On behalf of co-authors, we thank you very much for giving us an opportunity to revise our manuscript, we appreciate you very much for your positive and constructive comments and suggestions on our manuscript. We are also very grateful to you for your reference. In the future research, we will be more rigorous and careful.

Reviewer 2 Report

- In line 58-59, page 2, the authors mention that «Cui and Huang (2018) [16] discussed several schemes for raising the public finance of Green Climate Fund……». Please, be more specific, what kind of schemes?

-In my opinion the sub-section 2.2. (Interactive relationship between GF and GE) it is better to transferred as 2.1. and the sub-section 2.1 (Index system) to take the place of 2.2 sub-section.

-In Discussion and Conclusions section it is necessary to be indicating the impact from research results to the society, policies and the science and the further research that could be following on this object.

- In Discussion and Conclusions section someone can find concentrated the results of research for one more time, but without the necessary discussion about them.

-In Discussion and Conclusions section, authors are mentioning in two points  (562 and 585 line), that «this study recommends a high-level coordination organization that establishes economic development planning and environmental protection strategies from a transregional perspective……AND this paper is of practical significance for enlightening the regional development policy of China» BUT these real hot points not analyzed more. The study is ending (in my opinion) to the methodological role of the proposed evaluation system of coupling between green finance and green economy without to propose or to suggest something practical.

-In Discussion or Conclusions section there is no any mention about the limitations of the study and the limitations for the proposed evaluation procedure also between GF and GE.

- When you use an abbreviation for first time it is necessary to give their full explanation (even if the abbreviation is well-known, please make the correction).

-Please add the sources below the Tables (or in the titles of them) where are needed.

-The research results are mentioning exclusively to the China reality, what about the implementation, the comparison and the proposals for other countries and especially Western countries in which the GF and GE is in another stage and level?

Author Response

Response to Reviewer

Nana Liua, Chuanzhe Liua*, Yufei Xiab, Yi Rena , Jinzhi Lianga

At first, we appreciate editors’ and reviewers’ valuable comments and suggestions which help us improve the paper significantly.

Response to Reviewer 2 Comments

Point 1: In line 58-59, page 2, the authors mention that «Cui and Huang (2018) [16] discussed several schemes for raising the public finance of Green Climate Fund……». Please, be more specific, what kind of schemes?

Response1: Thanks for the suggestion. We have added five schemes for raising the public finance of Green Climate Fund discussed by Cui and Huang (2018).  “Namely, the historical emission responsibility (HR), ability to pay (AP), United Nations (UN) membership dues, the Official Development Assistance (ODA), and the Global Environment Facility (GEF) approaches. Among these schemes, HR and AP have been widely examined, whereas the remaining three schemes draw lessons from ongoing international financing mechanisms.”

Point 2: In my opinion the sub-section 2.2. (Interactive relationship between GF and GE) it is better to transferred as 2.1. and the sub-section 2.1 (Index system) to take the place of 2.2 sub-section.

Response2: Thanks for the suggestion. We have transferred sub-section 2.2. (Interactive relationship between GF and GE) as 2.1 and the sub-section 2.1 (Index system) taken the place of 2.2.

Point 3: In Discussion and Conclusions section it is necessary to be indicating the impact from research results to the society, policies and the science and the further research that could be following on this object.

Response3: Your opinion is very important, and we greatly appreciate your feedback. We have added the implication from our research results for the society, polices, government and science.

This paper may contribute to the realization of social and economic sustainable development through exploring the coronation between green finance and green economy. 

Our results provide a fact that there is a certain gap in the coordinated development of green finance and green economy for 30 provinces in China. Therefore, in order to further narrow this gap, it is necessary to strengthen the construction of a diversified green financial system in the central and western regions, for example, vigorously develop green bonds, energy conservation and environmental protection risk investment funds, and introduce clean development trading projects and avoid the transfer of polluting industries to the central and western regions, especially in areas with weak environmental and ecological carrying capacity, so as to achieve comprehensive coordination between regions and promote green finance and green economy and achieve the country's overall social sustainable development.

In addition, this paper also provides some suggestions for government about how to effectively execute the financial policies. Specifically, the policymakers need to recognize the coupling and coordination between green finance and the green economy. It is important not to focus only on changes in the scale of the one, but to make it a policy objective to promote coordination development between the two. The first is to improve the investment channels of the green financial system and raise the level of investment; the second is to improve the relevant policies and provide policy support for the development of green industries in order to promote the balanced development of  the two systems. Also, the regulator should establish a monitoring and feedback system between the two systems and enhance the operational efficiency of the coupled system through a positive feedback effect. It is important to establish and improve a system for monitoring the use of funds by enterprises, earmarking funds for green enterprises and improving the efficiency of their use. A feedback mechanism should also be improved to fully receive feedback on the effectiveness of the use of green finance policies. Policies already enacted should be revised based on market feedback to better promote green industries. At the same time, green financial institutions should be guided to establish information service platforms, strengthen exchanges with green enterprises, better understand the financing needs of energy-saving and environmental protection enterprises and provide them with matching financing solutions.

It is interesting and important that our application of the Spatial Markov approach to the field of environmental finance, in particular to the coordinated development of GF and GE. We provide new ideas and insights for enriching the research literature on green finance and green economy and coordinated development.

Therefore, in order to better achieve the coordinated development of green finance and green economy, we will further explore the path of green finance for green economy and the dynamic coordination between the two.”

Point 4: In Discussion and Conclusions section someone can find concentrated the results of research for one more time, but without the necessary discussion about them.

Response4: Thanks for your advice. We have deleted two paragraphs about the detail research results in Discussion and Conclusion.

 “in the GE system,…, For GF system, green credit plays an important role” , and ”The space Markov transfer probability matrix P,…, GF and GE shows the HH and LL clustering patterns in the long run.” respectively.

Point 5: In Discussion and Conclusions section, authors are mentioning in two points  (562 and 585 line), that «this study recommends a high-level coordination organization that establishes economic development planning and environmental protection strategies from a transregional perspective……AND this paper is of practical significance for enlightening the regional development policy of China» BUT these real hot points not analyzed more. The study is ending (in my opinion) to the methodological role of the proposed evaluation system of coupling between green finance and green economy without to propose or to suggest something practical.

Response5: Thanks for your advice, and we think your advice is very important. Referring to your previous comments. (point 3), we have changed and added the implication from our research results for the society, polices, government and science. Please see responds 3.

Point 6: In Discussion or Conclusions section there is no any mention about the limitations of the study and the limitations for the proposed evaluation procedure also between GF and GE.

Response6: Thanks for your suggestion. We have added several certain limitations at the end of the Discussion and Conclusion.

“However, this study has certain limitations. First, in terms of the construction of green finance indicators, generally, the green financial system can be divided into five areas, namely green credit, green securities, green insurance, green investment and carbon finance. But this paper does not include carbon finance in the indicator system due to a serious lack of data, even if a small percentage of the development of green finance in the carbon finance. Second, this paper evaluates the spatial distribution difference and dynamic evolution trend of the coordination by introducing global/local spatial autocorrelation, space Markov chain and local indicators of spatial association (LISA) Markov chain. However, considering the accuracy of results, we only predict the coordination in the short term, but not in the long term (e.g., over the next 10, 20 years). In the future, we hope to take a more appropriate approach to predicting the long-term degree of the coordination.

Third, this paper predicts the degree of coordination between GF and GE, but there is no further research on the early warning model of the coordination. Therefore, in the future we will explore the early warning model of coordinated development of GF and GE to provide better policy recommendations for sustainable development.

Point 7: When you use an abbreviation for first time it is necessary to give their full explanation (even if the abbreviation is well-known, please make the correction).

Response7: Thanks for your suggestion. We have added a full explanation for all abbreviations in this article.

Point 8: Please add the sources below the Tables (or in the titles of them) where are needed.

Response8: Thanks for your suggestion. We have added data sources below Tables 1 and 2, respectively.

Point 9: The research results are mentioning exclusively to the China reality, what about the implementation, the comparison and the proposals for other countries and especially Western countries in which the GF and GE is in another stage and level?

Response9: Thanks for your advice, and we think your advice is very important. For a better comparison with China, we have added the case of the United States, the European Union (EU) and some emerging economies in green finance development, and some suggestions learned from their implementation.

“Some developed countries are more mature in green finance and green economy development and may provide some experience to China. For example, the United States, as a developed economy, had an early start in green finance. Since the 1970s, the U.S. Congress has passed more than 20 laws on environmental protection related to water environment, air pollution, waste management, cleanup of polluted sites, etc. In 1980, the U.S. federal government introduced the Comprehensive Environmental Response, Compensation and Liability Act, which makes banks responsible for environmental pollution caused by their customers. In addition, the European Union (EU) attaches great importance to the development of green finance, with a mature legal system and active product innovation. The EU incentivizes green projects through tax incentives, government guarantees. For example, the German Government gives certain subsidies and interest rates for loans for green projects; the European Union provides tax incentives for green credits and securitized products; the British Government uses a "loan guarantee scheme" to support small and medium-sized enterprises (SMEs), especially environmental SMEs, etc.

In recent years, green finance has developed rapidly in emerging economy countries. The Central Bank of Brazil introduced a new regulatory approach in April 2014 that requires commercial banks to develop strategic actions and governance frameworks for environmental and social risk management and to implement them as core elements of overall risk management, and currently about 10 per cent of bank loans in Brazil are classified as green loans. In particular, the Bangladesh Central Bank has made increasing financial inclusion an explicit objective of the central bank's monetary policy and has provided credit guidelines for commercial banks that include new energy, pollution control, energy efficiency, etc., and the share of green credit now stands at 5 percent.

Compared to these countries discussed above, there are a number of problems and challenges in green finance development for China, such as: a) the overall proportion of investment and loans for green projects by financial institutions is still low. b) the lack of a green financial indicator system and incentive mechanisms. c) the Government's policy of supporting green industries is inadequate. d) the failure to establish a virtuous circular market mechanism for building ecological civilization. Therefore, it is necessary to learn from the experience of some developed countries for China, further improve the construction of green finance laws and regulations, insist on market-based operation, and encourage market innovation as a long-term mechanism for optimizing the allocation of green finance funds, enhancing the efficiency of the use of green finance funds and building China's green finance development.”

On behalf of co-authors, we thank you very much for giving us an opportunity to revise our manuscript, we appreciate you very much for your positive and constructive comments and suggestions on our manuscript. We are also very grateful to you for your reference. In the future research, we will be more rigorous and careful.
